# Diagnostic accuracy of digital technologies compared with 12-lead ECG in the diagnosis of atrial fibrillation in adults: A protocol for a systematic review

**Vethanayagam Antony Sheron**[1], **Rajendra Surenthirakumaran**[1], **Tiffany E. Gooden**[2],
**Gregory Y. H. Lip**[2,3,4], **G. Neil Thomas**[2]*, **David J. Moore**[2], **Krishnarajah Nirantharakumar**[2],
**Balachandran Kumarendran**[1,2]*, **Kumaran Subaschandran**[1],
**Shribavan Kanesamoorthy**[1], **Powsiga Uruthirakumar**[1], **Mahesan Guruparan**[5], on behalf of
the NIHR Global Health Research Group on Atrial Fibrillation Management[¶]

1 Faculty of Medicine, Department of Community and Family Medicine, University of Jaffna, Jaffna, Sri
Lanka, 2 Institute of Applied Health Research, University of Birmingham, Birmingham, United Kingdom,
3 Liverpool Centre for Cardiovascular Science at University of Liverpool, Liverpool John Moores University
and Liverpool Heart & Chest Hospital, Liverpool, United Kingdom, 4 Department of Clinical Medicine, Danish
Center for Health Services Research, Aalborg University, Aalborg, Denmark, 5 Department of Cardiology,
Teaching Hospital, Jaffna, Sri Lanka

¶ Members of the NIHR Global Health Research Group on Atrial Fibrillation Management are listed in the
Acknowledgments
* b.kumarendran@bham.ac.uk (BK); G.N.Thomas@bham.ac.uk (GNT)

pone.0301729

Universita degli Studi di Siena, ITALY

**Data Availability Statement:** No datasets were
generated or analysed during the current study. All

## Abstract

### Background

Atrial fibrillation (AF) is the most prevalent cardiac arrhythmia in the world. AF increases the
risk of stroke 5-fold, though the risk can be reduced with appropriate treatment. Therefore,
early diagnosis is imperative but remains a global challenge. In low-and middle-income
countries (LMICs), a lack of diagnostic equipment and under-resourced healthcare systems
generate further barriers. The rapid development of digital technologies that are capable of
diagnosing AF remotely and cost-effectively could prove beneficial for LMICs. However, evi-
dence is lacking on what digital technologies exist and how they compare in regards to diag-
nostic accuracy. We aim to systematically review the diagnostic accuracy of all digital
technologies capable of AF diagnosis.

### Methods

MEDLINE, Embase and Web of Science will be searched for eligible studies. Free text
terms will be combined with corresponding index terms where available and searches will
not be limited by language nor time of publication. Cohort or cross-sectional studies com-
prising adult (≥18 years) participants will be included. Only studies that use a 12-lead ECG
as the reference test (comparator) and report outcomes of sensitivity, specificity, the diag-
nostic odds ratio (DOR) or the positive and negative predictive value (PPV and NPV) will be
included (or if they provide sufficient data to calculate these outcomes). Two reviewers will

relevant data from this study will be made available upon study completion.

**Funding:** This research was funded by the (National Institute of Health Research) NIHR (17/63/121) using UK aid from the UK Government to support global health research. The views expressed in this publication are those of the author(s) and not necessarily those of the NIHR or the UK Department of Health and Social Care. Author who received fund - GNT (G. Neil Thomas) - The funder had no role in the design of the study or the writing of this protocol. The funder will not have a role in the data synthesis for this overview or in the writing of the manuscript. NIHR, UK - https://www.nihr.ac.uk/.

**Competing interests:** The authors declare that they have no competing interests.

**Abbreviations:** AF, atrial fibrillation; LMICs, low- and middle-income countries; apps, smart applications; mHealth, mobile health; eHealth, electronic health; ECG, electrocardiogram; PPG, photoplethysmography; PROSPERO, International Prospective Register of Systematic Reviews; PRISMA-P, Preferred Reporting Items for Systematic Reviews and Meta-Analysis Protocols; DOR, diagnostic odds ratio; NPV, negative predictive value; PPV, positive predictive value; TP, true positive; TN, true negative; FP, false positive; FN, false negative; CENTRAL, Cochrane Central Register of Controlled Trials; QUADAS-2, Quality Assessment of Diagnostic Accuracy Studies tool; RoB 2, Risk of Bias 2.0; ROBINS-I, Risk of Bias in Non-randomized studies of interventions; SROC, summary receiver-operating curve; GRADE, Grading of Recommendations Assessment, Development and Evaluation.

independently assess articles for inclusion, extract data using a piloted tool and assess risk of bias using the QUADAS-2 tool. The feasibility of a meta-analysis will be determined by assessing heterogeneity across the studies, grouped by index device, diagnostic threshold and setting. If a meta-analysis is feasible for any index device, pooled sensitivity and specificity will be calculated using a random effect model and presented in forest plots.

## Discussion

The findings of our review will provide a comprehensive synthesis of the diagnostic accuracy of available digital technologies capable for diagnosing AF. Thus, this review will aid in the identification of which devices could be further trialed and implemented, particularly in a LMIC setting, to improve the early diagnosis of AF.

## Trial registration

### Systematic review registration:

PROSPERO registration number is CRD42021290542. https://www.crd.york.ac.uk/prospero/display_record.php?ID=CRD42021290542.

## Background

Atrial fibrillation (AF) is the most common cardiac arrhythmia in the world with prevalence predicted to double over the next 30 years [1]. The risk of stroke is five-fold in AF patients; however, early anticoagulation therapy can reduce this risk by 65% [2]. Early diagnosis of AF is key for ensuring early effective treatment thus avoiding fatal and debilitating strokes. It is estimated that 15% of people with AF in high-income countries are not diagnosed, of whom up to 75% could be eligible for anticoagulation therapy [3]. In low- and middle-income countries (LMICs), under diagnosis is amplified due to limited resources, lack of knowledge of AF (in primary care and in the public) and cost implications [4,5].

Digital health technologies to improve the accuracy and diagnostic rate of AF have rapidly advanced in the last decade [6,7]. Examples include smart watches, smart belts, single lead mobile based electrocardiogram (ECG), portable Holter monitoring devices, modified blood pressure monitors and photoplethysmography (PPG) [7]. These technological advancements are often minimally-intrusive and enable self-testing and ad-hoc long-term monitoring [6]; thus, improving early diagnosis and self-monitoring by mitigating barriers of inaccessibility to and non-attendance at healthcare facilities. Advanced devices allow for the transfer of physiological parameters and patient-reported symptoms directly from smartphone applications to healthcare providers, reducing the number of healthcare visits necessary; thus, reducing costs on the patient and healthcare system [6,8]. Mobile connectivity has exponentially increased globally over the last 20 years, including in LMICs where now more than half the population use mobile internet [9,10]. As the growth of mobile connectivity continues to rise, the benefits of using digital technologies will become increasingly notable in LMICs where AF burden is increasing the most [11].

Eight systematic reviews were identified on the diagnostic accuracy of digital technologies such PPG, mobile applications, single lead ECGs and smart wearable devices for AF [12–20] However, six of these reviews only assessed one type of technology. For instance, Belani et al (2021), Gill S et al (2022) [12,17] Nazarian S (2020) et al and Narut et al (2021) only reviewed

the diagnostic accuracy of smart gadgets and wearable devices such smart watches [13,18]; and Wong et al (2020) only reviewed diagnostic accuracy of handheld electrocardiogram [20]; and O'Sullivan JW et al reviewed diagnostic accuracy of smartphone camera applications [16]. Including all types of technologies in one review, with pooled diagnostic accuracy stratified by type of device would be useful in deciding which device to take forward for testing in new settings, particular in resource-limited settings. Giebel et al reviewed the diagnostic accuracy of all types of technologies; however, the inclusion criteria were restricted to publication years 2014 to 2019, they did not restrict to studies that only used a 12-lead ECG (the gold standard) as the comparison device. Using different comparison devices can lead to discrepancies in diagnostic accuracy. Lopez et al also reviewed all types of technologies, but also limited their search to 2012 to 2019 and did not restrict to studies using the gold standard as a comparator device (including several studies where the comparator was not described). Furthermore, several studies that report on the AF diagnostic accuracy of digital technologies have been published since the search was conducted for these two existing systematic reviews. Thus, an updated comprehensive review on the diagnostic accuracy of all digital technologies compared to the gold standard will aid in the selection of which digital device(s) can and should be tested in LMICs. Therefore, we aimed to conduct a systematic review to assess the diagnostic accuracy of digital technologies that can be used for diagnosing AF. To reach this aim, we propose to answer the following questions:

Q1: What digital technologies has available diagnostic accuracy data for AF diagnosis?

Q2: What digital technology is the most accurate for diagnosing AF?

## Methods

This protocol has been registered within the International Prospective Register of Systematic Reviews (PROSPERO) database [21] (registration number CRD42021290542) and is reported in line with the Preferred Reporting Items for Systematic Reviews and Meta-Analysis Protocols (PRISMA-P) (S1 File) [22]. The methods of this systematic review is in line with the Cochrane Handbook for Systematic Reviews [23].

### Eligibility criteria

Study eligibility will be determined through a pre-defined PICOS criteria (Table 1).

**Table 1. PICOS of the systematic review.**

| PICOS | Inclusion criteria | Exclusion criteria |
|---|---|---|
| Population | Adults aged 18 years or above | |
| Intervention | Mobile application, ECG based digital technology or non-ECG-based digital technology | |
| Comparison | 12 lead ECG | |
| Outcome | Sensitivity and specificity, the diagnostic odds ratio (DOR), negative and positive predictive values (NPV and PPV), or true positive (TP), true negative (TN), false positive (FP) and false negative (FN) values for detecting atrial fibrillation | Studies that do not reporting any of these measures for atrial fibrillation will be excluded |
| Study design | Cohort and cross-sectional study designs | Interventional studies, review articles, editorials, case reports, modelling, economic studies, or clinical trials |

**Inclusion criteria.** We will include primary studies that assess AF within an adult participants aged 18 years or older (*population*) using any mobile application, ECG based digital technology or non-ECG-based digital technology (*intervention*) compared with a 12-lead ECG (*comparator*) and reporting on the following *outcomes* sensitivity and specificity, the diagnostic odds ratio (DOR), negative and positive predictive values (NPV and PPV), or figures that can be used by review authors to calculate these measures: (i.e. true positive (TP), true negative (TN), false positive (FP), false negative (FN). Only cohort and cross-sectional *study designs* will be deemed eligible.

Mobile applications that will be considered includes but are not limited to Cardiogram application and Cardio Rhythm. ECG based digital technologies that will be considered includes but are not limited to handheld devices, smart watches, single lead ECG (Kardia), wearables (bio-textiles, belt) and patches. Non-ECG based digital technologies that will be considered includes but are not limited to PPG, oscillometry, mechanocardiograph, modified BP meters and contactless video PPG.

Authors will calculate the outcomes of interest from studies that only report true positive and negative values using the following:

- TP/(TP+FN) = Sensitivity

- TN/(TN+FP) = Specificity

- (TP/FN)/(FP/TN) = DOR

- TN/(TN+FN) = NPV

- TP/(TP+FP) = PPV

**Inclusion criteria.** We will exclude any studies that do not report the outcomes of interest specifically for AF and the following study designs: interventional studies, review articles, editorials, case reports, modelling, economic studies, or clinical trials.

## Search strategy

MEDLINE and Embase via OVID will be searched for eligible studies. A search of grey literature will be conducted using Web of Science database. References of all included studies and existing systematic reviews identified from the search will be assessed for additional eligible studies not identified within the search.

Key terms such as "atrial fibrillation", "mHealth", "eHealth", "sensitivity" and "specificity" will be used alongside MeSH and Emtree terms where appropriate. A draft search strategy for MEDLINE is available (S2 File). There will be no restriction on publication date, setting or language. Citations of all identified studies from the search strategy will be exported to EndNote X9 software (Clarivate 2013) and duplicates will be removed using the automated features. Two reviewers (AS and SK) will independently review titles and abstracts and the full-text of any study where eligibility cannot be determined via the title and abstract. Disagreements will be handled through discussion or a third reviewer (UP).

## Data extraction and management

Relevant data will be extracted from included studies using a piloted extraction tool in Excel. Data will be extracted by two reviewers (AS and SK) independently and cross checked for accuracy. We will contact the study authors in cases where additional information is required.

If there is no response after two contact attempts, we will continue with the data available within the published article.

The following information will be recorded:

- Study characteristics (title, authors, journal, publication date, study period, number of participants, country, any conflicts of interest, funding source)

- Study design/methodology (study type, data source/recruitment strategy, eligibility criteria, data analysis)

- Study population characteristics (age, sex, number of suspected AF cases)

- Index test(s) (type of digital technology) used

- Reference test(s) used

- Outcome (sensitivity and specificity, DOR, PPV, NPV, TP, TN, FP and FN for each mHealth being tested and for each test duration presented)

## Risk of bias assessment

The Quality Assessment of Diagnostic Accuracy Studies tool (QUADAS-2) will be used to assess risk of bias independently by two reviewers (AS and SK) [24]. The checklist includes questions on patient selection, index test, reference standard and flow and timing; answers will be noted as yes, no or unclear for signaling questions and high, low or unclear for risk of bias and applicability questions. Bias will not be assessed using one overall score; instead, each question will be tabulated for each study and will be considered within the discussion of our results. The QUADAS-2 tool will be piloted as is and where necessary, the signaling questions will be modified to fit the purpose of our review. Once implemented, any disagreements will be resolved by discussion or a third reviewer (UP).

## Analysis

The study selection process will be summarized using a PRISMA diagram [25]. Tables summarizing the study characteristics of included studies, risk of bias (RoB) and quality of evidence will be presented in a summary of findings table.

Excel will be used for data management and R 4.1.1 (R Core Team, 2020) will be used to perform the statistical analysis. We will present the outcomes of sensitivity and specificity using forest plots and will group the studies based on index test first, then by diagnostic threshold of the index test, then by country setting (high-income or low/middle-income) and setting of application (community or healthcare facility). If more than one primary study has the same index test and diagnostic thresholds, heterogeneity will be assessed through the presented forest plots. If there are more than one primary study with the same index test but differing thresholds, heterogeneity will be assessed by observing how well studies fit within the plotted summary receiver operating curve (SROC). If heterogeneity is suspected, a Chi-squared test will be performed. Heterogeneity based on sample size ($\leq$100 participants and >100 participants), sampling method (consecutive/convenience), country setting (high-income vs low/middle-income country) and setting of application (community and healthcare facility) will be explored if necessary. If heterogeneity exists but cannot be explained, a meta-analysis will not be performed and results from included studies will be qualitatively synthesized. Where a meta-analysis is possible for any group of primary studies with the same index test, the sensitivity and specificity will be calculated using a random effect model and pooled effects will be presented within the forest plot of individual study estimates.

Confidence in effect estimates for the outcome measured will be rated according to the quality of evidence using Grading of Recommendation, Assessment, Development and Evaluation (GRADE) approach for diagnostic tests and strategies [26].

## Discussion

AF is the most common cardiac arrhythmia worldwide and a major contributor to the global burden of stroke [27]. Early diagnosis of AF can prevent the occurrence of stroke and reduce excess morbidity and mortality [28]. Whilst screening programmes have been found to be clinically effective [29], the 12-lead ECG (gold-standard diagnostic tool for AF) must be used and assessed by a trained clinician within a healthcare facility and can be costly to use in a population-based screening programme [3]. Given 12-lead ECGs are not portable; patients must travel to a healthcare facility to be assessed which is not always accessible or practical and can be costly for the patient and healthcare system, particularly in LMICs where access to healthcare facilities and resources are limited [5,30]. The continuous Holter monitoring device is a useful alternative and is currently the method most used globally to detect suspected AF; however, they are expensive and not commonly available in LMICs. Furthermore, it can be a burden for the patient to carry the device for one or two days [19].

To date, a number of digital technologies have been developed that can be used to diagnose AF, with the added benefits of being portable, patient-friendly and cost effective [10]. Such devices could be particularly beneficial in diagnosing AF in at-risk populations in LMICs where the lack of healthcare access and diagnostic equipment are a known barrier for early AF diagnosis [31]. An improved understanding of what devices have AF diagnostic accuracy data available, and which provide higher accuracy will aid the selection of which devices to further test and use in LMIC settings. Our review findings will therefore provide a clinically important resource for LMICs, where the prevalence of AF is expected to exponentially increase over the next decade [4].

We are only including studies that compare the diagnostic accuracy of digital technologies with a 12-lead ECG; therefore, we may unintentionally exclude many studies that report the diagnostic accuracy of digital technologies, but not against this gold-standard. However, from assessing existing systematic reviews [12,13,15,19,32,33], we do not expect this eligibility criteria to exclude a large number of studies. Furthermore, we feel it is necessary to compare the diagnostic accuracy of these devices to the 12-lead ECG for comparison purposes and to fully understand the potential of using these devices for diagnosing AF. Off-brand named devices may be missed within our search; however, we expect from our search of numerous databases and assessment of references from included studies and existing systematic reviews, the number of studies missed will be reduced. We expect that the methodological quality may be lower for most studies given the observational and descriptive nature of the study designs. Nonetheless, a comprehensive review of all available evidence will be a vital step in understanding what device is best to test and use for diagnosing AF, particularly in low-resourced areas of the world. We will use the findings of our risk and bias assessment to critically assess and synthesize the evidence in a clear and precise manner.

Publication of this research protocol is in keeping with transparent research practice. If any methods outlined in this protocol changes substantially following publication, we will document the amendments. Dissemination of the findings from this review will be done through presenting at academic seminars and conferences and submitting a manuscript to a peer-reviewed journal. This review will provide guidance to a range of stakeholders such as clinicians, primary care doctors and policy makers on the selection of appropriate technology to test and use for early diagnosis of AF.

## Supporting information

**S1 File. PRISMA-P checklist.**
(DOCX)

**S2 File. Search strategy for MEDLINE.**
(DOCX)

## Acknowledgments

Members of the **NIHR Global Health Group on Atrial Fibrillation Management** listed in alphabetical order: Ajini Arasalingam, Abi Beane, Isabela M Bensenor, Peter Brocklehurst, Kar Keung Cheng, Itamar S Santos, Wahbi El-Bouri, Mei Feng, Alessandra C Goulart, Sheila Greenfield, Yutao Guo, Mahesan Guruparan, Gustavo Gusso, Tiffany E Gooden, Rashan Haniffa, Lindsey Humphreys, Kate Jolly, Sue Jowett, Balachandran Kumarendran, Emma Lancashire, Deirdre A Lane, Xuewen Li, Gregory Y.H. Lip (Co-PI), Yan-guang Li, Trudie Lobban, Paulo A Lotufo, Semira Manseki-Holland, David J Moore, Krishnarajah Nirantharakumar, Rodrigo D Olmos, Elisabete Paschoal, Paskaran Pirasanth, Uruthirakumar Powsiga, Carla Romagnolli, Alena Shantsila, Vethanayagam Antony Sheron, Kanesamoorthy Shribavan, Isabelle Szmigin, Kumaran Subaschandren, Rajendra Surenthirakumaran, Meihui Tai, Bamini Thavarajah, G. Neil Thomas (Co-PI) Ana C Varella, Hao Wang, Jingya Wang, Hui Zhang, Jiaoyue Zhong.

## Author Contributions

**Conceptualization:** Vethanayagam Antony Sheron, Rajendra Surenthirakumaran, Tiffany E. Gooden, Gregory Y. H. Lip, G. Neil Thomas, David J. Moore, Krishnarajah Nirantharakumar, Balachandran Kumarendran, Kumaran Subaschandren, Shribavan Kanesamoorthy, Powsiga Uruthirakumar, Mahesan Guruparan.

**Supervision:** Rajendra Surenthirakumaran, Gregory Y. H. Lip, G. Neil Thomas, David J. Moore, Krishnarajah Nirantharakumar, Balachandran Kumarendran, Kumaran Subaschandran, Mahesan Guruparan.

**Validation:** G. Neil Thomas, David J. Moore.

**Writing – original draft:** Vethanayagam Antony Sheron, Shribavan Kanesamoorthy, Powsiga Uruthirakumar.

**Writing – review & editing:** Rajendra Surenthirakumaran, Tiffany E. Gooden, G. Neil Thomas, David J. Moore, Balachandran Kumarendran.

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
