## [Decision Letter · Decision Letter 0]

27 Mar 2023

PONE-D-23-03001

Diagnostic accuracy of digital technologies compared with 12-lead ECG in the diagnosis of atrial fibrillation in adults: A protocol for a systematic review

PLOS ONE

Dear Dr. Vethanayagam,

Thank you for submitting your manuscript to PLOS ONE. After careful consideration, we have decided that your manuscript does not meet our criteria for publication and must therefore be rejected.

I am sorry that we cannot be more positive on this occasion, but hope that you appreciate the reasons for this decision.

Kind regards,

Burak Katipoğlu

Academic Editor

PLOS ONE

Additional Editor Comments:

Dear Authors, I regret to inform you that the language and sentence structures of this manuscript are at times incomprehensible. The paper needs rewriting and thorough language editing to allow for a proper peer review.

First of all, I think there are too many subjective sentences in the introduction part of the article.

I could not find how many articles were included in this systematic review. And I couldn't find how many articles were excluded and why.

I could not find any evidence in the references that Digital Technologies are more cost-effective than diagnosis with a 12-lead ECG. Is AF recognition training more expensive in 12-lead ECG?

Mobile devices are used a lot in the world, but can these phones be less accessible in low and middle income countries? It has not been reported which of the digital technologies determined as the primary outcome can be used in these countries.

I could not conclude which device is the most diagnostic as a result of this article.

There is no forest plot in the tables section.

In this systematic review, the methodology seems to have been described but not concluded.

In its current state, I do not recommend accepting this paper.

Reviewers' comments:

Reviewer's Responses to Questions

**Comments to the Author**

1. Does the manuscript provide a valid rationale for the proposed study, with clearly identified and justified research questions?

Reviewer #1: Yes

Reviewer #2: Yes

Reviewer #3: Partly

Reviewer #4: Partly

2. Is the protocol technically sound and planned in a manner that will lead to a meaningful outcome and allow testing the stated hypotheses?

Reviewer #1: Partly

Reviewer #2: Yes

Reviewer #3: Partly

Reviewer #4: Yes

3. Is the methodology feasible and described in sufficient detail to allow the work to be replicable?

Reviewer #1: Yes

Reviewer #2: Yes

Reviewer #3: Yes

Reviewer #4: Yes

4. Have the authors described where all data underlying the findings will be made available when the study is complete?

Reviewer #1: Yes

Reviewer #2: No

Reviewer #3: No

Reviewer #4: No

5. Is the manuscript presented in an intelligible fashion and written in standard English?

Reviewer #1: No

Reviewer #2: Yes

Reviewer #3: Yes

Reviewer #4: No

6. Review Comments to the Author

You may also provide optional suggestions and comments to authors that they might find helpful in planning their study.

Reviewer #1: I think the subject of the article is quite original and interesting. When I read the topic, it really excited me. But I think some definitions are missing. I couldn't understand how and where to use predictive values. Since I do not know the number of data in the study, I cannot comment on this issue. At the same time, the exclusion and inclusion criteria in the study were not clearly explained. Besides, I think that the English used makes it difficult to understand. I think it will be a good interesting article when these are edited.

Reviewer #2: After reviewing the current manuscript, our comment is regarding:

The inclusion and exclusion criteria should be clear. And better to add an illustration that would demonstrate the protocol and proposed flow of the systematic review.

Regards

Reviewer #3: Dear Editor, Thank you for giving us the opportunity to review this article titled

“Diagnostic accuracy of digital technologies compared with 12-lead ECG in the diagnosis of

atrial fibrillation in adults: A protocol for a systematic review”.

The subject mentioned in the article is as if it was just designed and just remained. We see that the diagnostic method compared methodologically is not standardized. smart watch, smart belts, holter etc. We could not find any data on which of these devices is more effective in the old literature in which the normal technique was compared. The prevalence of smartphone use has been mentioned, but we see that there is no need for additional devices to smartphones for this diagnostic method and data on how widely these devices are used are not mentioned. We cannot see an analysis of the cost of the additional devices to be given to each individual and the evaluation made by a trained person in the study. In addition, this situation is a situation that may be more appropriate in high-income societies, not in low and middle-income countries. Again, we cannot see the values of the results obtained from the study list included in the meta-analysis in the study text. This obscures the issue of how the study was designed and how the data was obtained and how it was interpreted. This creates the opinion that the evaluation is based on a wish, not on objective parameters. The article is not suitable for publication as it is.

Reviewer #4: Dear Authors, Thank you for giving us the opportunity to review this article titled “Diagnostic accuracy of digital technologies compared with 12-lead ECG in the diagnosis of atrial fibrillation in adults: A protocol for a systematic review”.

I regret to inform you that the language and sentence structures of this manuscript are at times incomprehensible. The paper needs rewriting and thorough language editing to allow for a proper peer review.

First of all, I think there are too many subjective sentences in the introduction part of the article.

I could not find how many articles were included in this systematic review. And I couldn't find how many articles were excluded and why.

I could not find any evidence in the references that Digital Technologies are more cost-effective than diagnosis with a 12-lead ECG. Is AF recognition training more expensive in 12-lead ECG?

Mobile devices are used a lot in the world, but can these phones be less accessible in low and middle income countries? It has not been reported which of the digital technologies determined as the primary outcome can be used in these countries.

I could not conclude which device is the most diagnostic as a result of this article.

There is no forest plot in the tables section.

In this systematic review, the methodology seems to have been described but not concluded.

In its current state, I do not recommend accepting this paper.

Kind Regards.

7. PLOS authors have the option to publish the peer review history of their article (what does this mean?). If published, this will include your full peer review and any attached files.

Reviewer #1: **Yes: **Ayşe Büşra ÖZCAN

Reviewer #2: **Yes: **Rami Riziq Yousef Abumuaileq

Reviewer #3: No

Reviewer #4: No

- - - - -

---

## [Author Response · Author response to Decision Letter 0]

30 May 2023

Reviewer 1 comments:

Comment 1: I think the subject of the article is quite original and interesting. When I read the topic, it really excited me. But I think some definitions are missing.

Response 1: Thank you for the positive feedback. We hope you find our amendments and responses below have clarified any confusion around definitions. 

Comment 2: I couldn't understand how and where to use predictive values. 

Response 2: The negative and positive predictive values will either be extracted directly from the primary studies included in our systematic review or calculated if the study only provides the true negative and true positive values. We will then report and compare the predictive values for the digital devices included in our review. Please note that the outcome data (including the negative and positive predictive values) for each digital device reviewed will originate from studies where the diagnostic accuracy of the digital device has been compared to a 12 lead ECG (the gold standard).

Comment 3: Since I do not know the number of data in the study, I cannot comment on this issue. 

Response 3: Please note our manuscript is a protocol for the systematic review we plan to conduct; we therefore do not have the results of the systematic review to present. 

Comment 4: At the same time, the exclusion and inclusion criteria in the study were not clearly explained. 

Response 4: As per the Preferred Reporting Items for Systematic Reviews and Meta-Analysis Protocols (PRISMA-P), we have outlined the eligibility criteria in terms of the PICO on page 5-6, lines 108-115 and in Table 1. To summarise these criteria: we will include primary cohort and cross-sectional studies that include adults (≥18 years old) and report on the diagnostic accuracy of mobile applications, ECG-based digital technologies or non-ECG-based digital technologies compared to a 12-lead ECG. Only studies that report the following outcomes will be included: sensitivity and specificity, the diagnostic odds ratio (DOR), the negative and positive predictive values (NPV and PPV) or true positive and negative values. 

Comment 5: Besides, I think that the English used makes it difficult to understand. I think it will be a good interesting article when these are edited.

Response 5: We have thoroughly read the manuscript and have subsequently made amendments to improve the language and sentence structure throughout. We hope you will find these amendments have elevated our manuscript to publication standards. 

Reviewer 2 comments: 

Comment 1: The inclusion and exclusion criteria should be clear. 

Response 1: Please refer to response 4 to reviewer 1 above. 

Comment 2: And better to add an illustration that would demonstrate the protocol and proposed flow of the systematic review.

Response 2: We are unsure what kind of flow diagram you are suggesting we should add. Please note that we have mentioned in the ‘analysis’ section that we plan to present the PRISMA diagram of the study selection process in the final systematic review manuscript. However, if there is something more you think is needed at the protocol stage, please let us know and we will consider adding it in.

Reviewer 3 comments:

Comment 1: Dear Editor, Thank you for giving us the opportunity to review this article titled

“Diagnostic accuracy of digital technologies compared with 12-lead ECG in the diagnosis of

atrial fibrillation in adults: A protocol for a systematic review”. The subject mentioned in the article is as if it was just designed and just remained.

Response 1: Thank you. We are unsure what you mean by ‘the subject mentioned in the article is as if it was just designed and just remained.’ If this point could please be clarified, we can address it accordingly.

Comment 2: We see that the diagnostic method compared methodologically is not standardized. smart watch, smart belts, holter etc. We could not find any data on which of these devices is more effective in the old literature in which the normal technique was compared. 

Response 2: Apologies as we are unsure what you mean by ‘normal technique’; however, we will only include studies where the digital device has been compared with a 12-lead ECG (the gold standard) as per our eligibility criteria. From these studies, we will have the sensitivity, specificity, the diagnostic odds ratio (DOR) or the negative and positive predictive values (NPV and PPV) of each digital device to report and compare. 

Comment 3: The prevalence of smartphone use has been mentioned, but we see that there is no need for additional devices to smartphones for this diagnostic method and data on how widely these devices are used are not mentioned. 

Response 3: Please refer to response 4 to the editor above. 

Comment 4: We cannot see an analysis of the cost of the additional devices to be given to each individual and the evaluation made by a trained person in the study. 

Response 4: Please note our manuscript is a protocol for the systematic review we plan to conduct; we therefore do not have the results of the systematic review to present. We also do not plan to conduct a cost analysis as this is not within the scope of our systematic review. The aim of our systematic review is to assess the diagnostic accuracy of existing digital devices that can be used for diagnosing AF.

Comment 5: In addition, this situation is a situation that may be more appropriate in high-income societies, not in low and middle-income countries. 

Response 5: digital devices are increasingly being used in LMICs as they are often cost-effective and can overcome issues of healthcare access, particularly for rural inhabitants. As mentioned in our introduction (lines 72-82, as below), mobile access is steadily increasing in LMICs, which makes the topic of using digital technologies an increasingly important topic to explore for such settings. 

“These technological advancements are often minimally-intrusive and enable self-testing and ad-hoc long-term monitoring (6); thus, improving early diagnosis and self-monitoring by mitigating barriers of inaccessibility to and non-attendance at healthcare facilities. Advanced devices allow for the transfer of physiological parameters and patient-reported symptoms directly from smartphone applications to healthcare providers, reducing the number of healthcare visits necessary for the public; thus, reducing costs on the patient and healthcare system (6,8). Mobile connectivity has exponentially increased globally over the last 20 years, including in LMICs where now more than half the population use mobile internet (7,9). As the growth of mobile connectivity continues to rise, the benefits of using digital technologies will become increasingly notable in LMICs where AF burden is increasing the most (10).”

Comment 6: Again, we cannot see the values of the results obtained from the study list included in the meta-analysis in the study text. This obscures the issue of how the study was designed and how the data was obtained and how it was interpreted. This creates the opinion that the evaluation is based on a wish, not on objective parameters. 

Response 6: Please note our manuscript is a protocol for the systematic review we plan to conduct; we therefore do not have the results of the systematic review to present.

---

## [Decision Letter · Decision Letter 1]

2 Nov 2023

PONE-D-23-03001R1Diagnostic accuracy of digital technologies compared with 12-lead ECG in the diagnosis of atrial fibrillation in adults: A protocol for a systematic reviewPLOS ONE

Dear Dr. Kumarendran,

Thank you for submitting your manuscript to PLOS ONE. After careful consideration, we feel that it has merit but does not fully meet PLOS ONE’s publication criteria as it currently stands. Therefore, we invite you to submit a revised version of the manuscript that addresses the points raised during the review process. Please carefully address all the comments and concerns raised from the reviewers.

We look forward to receiving your revised manuscript.

Kind regards,

Ernesto Iadanza

Academic Editor

PLOS ONE

Reviewers' comments:

Reviewer's Responses to Questions

**Comments to the Author**

1. Does the manuscript provide a valid rationale for the proposed study, with clearly identified and justified research questions?

Reviewer #1: Yes

Reviewer #2: Yes

Reviewer #5: Partly

2. Is the protocol technically sound and planned in a manner that will lead to a meaningful outcome and allow testing the stated hypotheses?

Reviewer #1: Partly

Reviewer #2: Partly

Reviewer #5: Partly

3. Is the methodology feasible and described in sufficient detail to allow the work to be replicable?

Reviewer #1: Yes

Reviewer #2: Yes

Reviewer #5: Yes

4. Have the authors described where all data underlying the findings will be made available when the study is complete?

Reviewer #1: No

Reviewer #2: No

Reviewer #5: Yes

5. Is the manuscript presented in an intelligible fashion and written in standard English?

Reviewer #1: Yes

Reviewer #2: No

Reviewer #5: Yes

6. Review Comments to the Author

You may also provide optional suggestions and comments to authors that they might find helpful in planning their study.

Reviewer #1: Thank you for your revision. I think it is still not clear how and what kind of convenience the use of electronic devices can provide. However, the fact that the subject is new and interesting will guide different studies. Bets Regards..

Reviewer #2: The authors have partially addressed our comments.

A Research Protocol journal may fit better for the current manuscript.

Regards

Reviewer #5: This proposed systematic review aims to address an important health issue, atrial fibrillation, that is affecting 59.7 million people worldwide. It specifically mentions disparities in access to screening devices in low-middle-income countries (LMICs), which is an important gap in the current literature. If the authors consistently and cogently articulate this point throughout the manuscript, they would have made a strong case for their proposed review. Unfortunately, this point was not consistently carried through, as highlighted in my comments below.

1. The authors have highlighted that low-middle-income countries (LMICs) lack diagnostic equipment and are under-resourced in healthcare systems, generating further barriers. This is a good point. The costs of the screening devices could be barriers. Therefore, the authors may consider adding “device costs” as a parameter in their review. I have noted the authors' reply to Reviewer 3 that the device cost is outside the scope of the review. Paradoxically, the authors added, “As mentioned in our introduction (lines 72-82), mobile access is steadily increasing in LMICs, which makes the topic of using digital technologies an increasingly important topic to explore for such settings”. I perceive “not including device costs in the review” as a wasted opportunity to enhance the consistency and cogency of the authors’ argument that LMICs need affordable screening devices, and the review may generate useful information for LMICs. Of course, this matter is left to the decision of the authors and Editor-in-Chief.

2. The authors wrote, “No one review has reviewed the diagnostic accuracy of all existing devices, making it difficult to compare their accuracy.” This statement provokes arguments because (a) it is almost impossible to review all devices as there is a lack of standardisation and governance in publishing evaluation studies on devices, and (b) it is inappropriate to compare the accuracy of heterogeneous digital devices made of different technologies, such as comparing photoplethysmographic (PPG) devices, oscillometry, mechanocardiographic devices, modified BP meters versus ECG devices that produce rhythm traces. However, it would be more relevant and appropriate if the comparisons were stratified by type of technologies e.g., ECG, PPG, oscillometry etc. The authors can refer to the published systematic review and meta-analysis of the diagnostic accuracy of handheld ECG devices that produced rhythm traces in diagnosing AF (Wong KC, Klimis H, Lowres N, von Huben A, Marschner S, Chow CK. Diagnostic accuracy of handheld electrocardiogram devices in detecting atrial fibrillation in adults in community versus hospital settings: a systematic review and meta-analysis. Heart. 2020 Aug;106(16):1211-1217. doi: 10.1136/heartjnl-2020-316611) and “Giebel GD, Gissel C. Accuracy of mHealth Devices for Atrial Fibrillation Screening: Systematic Review; JMIR Mhealth Uhealth 2019;7(6):e13641”. In addition, the authors cited a series of systematic reviews and meta-analyses in their references 10 to 14. What are the gaps among these reviews? What else are you adding to the literature? The authors should explain how their proposed review addresses the gaps in the current literature.

3. Two reviewers commented about the lack of clarity in inclusion and exclusion criteria. The authors referred them to their existing statements in the manuscript. Perhaps the authors assume inclusion and exclusion criteria are mutually exclusive. However, the authors may think about other exclusion criteria, such as “studies that involved more than one person in interpreting the same ECG findings without reporting the consensus results among the interpreters.”

4. In the second comment, Reviewer 2 asked about the elaboration of the flow of review. The authors replied about including the PRISMA diagram, which is a conventional requirement. I think, in addition to the PRISMA diagram, the authors should elaborate on the number of reviewers in screening titles/ abstracts and how that flows into full-text review (e.g., X number of reviewers check the first N number of full text, resolve discrepancies & reach consensus, then Y number of reviewer proceed with the remaining full-text review).

5. The authors should be aware of potential spectrum effects that have been reported in the literature, i.e., variations of diagnostic accuracy due to differences in the settings in which the devices were used, such as in the community versus the hospital and characteristics of the users. A systematic review and meta-analysis of diagnostic accuracy of handheld ECG devices in diagnosing AF in community and hospital settings reported the variations in AF diagnostic accuracy (doi: 10.1136/heartjnl-2020-316611). Hence, the authors may consider adding the parameter "setting of application" in their review and explore the potential spectrum effect in their reviews.

7. PLOS authors have the option to publish the peer review history of their article (what does this mean?). If published, this will include your full peer review and any attached files.

Reviewer #1: No

Reviewer #2: **Yes: **Rami Riziq Yousef Abumuaileq

Reviewer #5: **Yes: **Kam Cheong Wong

---

## [Author Response · Author response to Decision Letter 1]

15 Feb 2024

15/02/2024

Dear Ernesto Iadanza, 

Manuscript ID number: PONE-D-23-03001R1

Thank you for reconsidering our manuscript titled “Diagnostic accuracy of digital technologies compared with 12-lead ECG in the diagnosis of atrial fibrillation in adults: A protocol for a systematic review”. We have addressed each comment in a point-by-point format below and have modified the manuscript accordingly. We have uploaded the updated manuscript with tracked changes. 

We would be greatly appreciative for your further consideration for publication in PLOS One.

Yours sincerely,

Dr B. Kumarendran 

Reviewer 1 comments:

Comment 1: Thank you for your revision. I think it is still not clear how and what kind of convenience the use of electronic devices can provide. However, the fact that the subject is new and interesting will guide different studies. 

Response 1: As described in the introduction section of the revised protocol, electronic devices are low cost compared with the existing medical diagnostic devices for atrial fibrillation (AF) (i.e. 12 lead ECG, ambulatory blood pressure monitors). Furthermore, electronic devices are portable and easily assessable and can be purchased even in low resourced settings for use in primary care or community-based settings which would be beneficial for low-and middle-income countries (LMICs). However, a comprehensive systematic review is needed to find out the hierarchy of diagnostic accuracy of existing electronic diagnostic devices for AF to inform future testing and trials in LMICs. 

Reviewer 2 comments: 

Comment 1: The authors have partially addressed our comments. A Research Protocol journal may fit better for the current manuscript.

Response 1: We feel we have comprehensively addressed each reviewers’ comments in the revised version of the manuscript; however, if there are any specific comments or suggestions for the protocol that we have not addressed, please kindly provide this feedback and we will consider amending the protocol further. PLOS One accepts study protocols as per their submission guidelines.

Reviewer 3 comments:

Comment 1: The authors have highlighted that low-middle-income countries (LMICs) lack diagnostic equipment and are under-resourced in healthcare systems, generating further barriers. This is a good point. The costs of the screening devices could be barriers. Therefore, the authors may consider adding “device costs” as a parameter in their review. I have noted the authors' reply to Reviewer 3 that the device cost is outside the scope of the review. Paradoxically, the authors added, “As mentioned in our introduction (lines 72-82), mobile access is steadily increasing in LMICs, which makes the topic of using digital technologies an increasingly important topic to explore for such settings”. I perceive “not including device costs in the review” as a wasted opportunity to enhance the consistency and cogency of the authors’ argument that LMICs need affordable screening devices, and the review may generate useful information for LMICs. Of course, this matter is left to the decision of the authors and Editor-in-Chief.

Response 1: We agree the costs of each device is an important factor for use in LMICs; however, costs of any given tool would be specific to a country/region and the use of the tool would be relative to specific average annual incomes to gauge affordability and this will vary even for the same type of tool. Given the nuances here, we will discuss the unit cost and costs implications and options for tools deemed to be of high diagnostic accuracy. 

Comment 2: The authors wrote, “No one review has reviewed the diagnostic accuracy of all existing devices, making it difficult to compare their accuracy.” This statement provokes arguments because (a) it is almost impossible to review all devices as there is a lack of standardisation and governance in publishing evaluation studies on devices, and (b) it is inappropriate to compare the accuracy of heterogeneous digital devices made of different technologies, such as comparing photoplethysmographic (PPG) devices, oscillometry, mechanocardiographic devices, modified BP meters versus ECG devices that produce rhythm traces. However, it would be more relevant and appropriate if the comparisons were stratified by type of technologies e.g., ECG, PPG, oscillometry etc. The authors can refer to the published systematic review and meta-analysis of the diagnostic accuracy of handheld ECG devices that produced rhythm traces in diagnosing AF (Wong KC, Klimis H, Lowres N, von Huben A, Marschner S, Chow CK. Diagnostic accuracy of handheld electrocardiogram devices in detecting atrial fibrillation in adults in community versus hospital settings: a systematic review and meta-analysis. Heart. 2020 Aug;106(16):1211-1217. doi: 10.1136/heartjnl-2020-316611) and “Giebel GD, Gissel C. Accuracy of mHealth Devices for Atrial Fibrillation Screening: Systematic Review; JMIR Mhealth Uhealth 2019;7(6):e13641”. 

Response 2: Thank you for the comments. To address point a: We agreed with the reviewer in practical difficulties of capturing all the range of different devices, but we tried to overcome this with subject experts and consultants in the digital technology of AF diagnosis (i.e – Lip G Y ,one of the world’s leading experts in risk stratification of atrial fibrillation, with his research directly informing the widely-used CHA₂DS₂-Vasc and HAS-BLED Scores and Author of 2020 ESC Guidelines for the digital diagnosis of AF), moreover, we have reviewed a list of commercially available digital technology devices on AF diagnosis. And we agree few reviews assessed multiple technologies in a single review, and we amended the gaps in those reviews in the introduction to clarify this point as follows: 

“Eight existing systematic reviews have assessed the diagnostic accuracy of digital technologies such as PPG, mobile applications, single lead ECGs and smart wearable devices for AF (12–20). However, six of these reviews only assessed one type of technology. For instance, Belani et al (2021), Gill S et al (2022), Nazarian S (2020) et al and Narut et al (2021) only reviewed the diagnostic accuracy of smart gadgets and wearable devices such smart watches (12, 13, 17, 18), and Wong et al (2020) only reviewed diagnostic accuracy of handheld electrocardiogram (20); and O'Sullivan JW (2020) et al reviewed diagnostic accuracy of smartphone camera applications (16). Including all types of technologies in one review, with pooled diagnostic accuracy stratified by type of device would be useful in deciding which device to take forward for testing in new settings, particular in resource-limited settings. Giebel et al reviewed the diagnostic accuracy of all types of technologies; however, the inclusion criteria were restricted to publication years 2014 to 2019, they did not restrict to studies that only used a 12-lead ECG (the gold standard) as the comparison device. Using different comparison devices can lead to discrepancies in diagnostic accuracy. Lopez et al also reviewed all types of technologies, but also limited their search to 2012 to 2019 and did not restrict to studies using the gold standard as a comparator device (including several studies where the comparator was not described). Furthermore, several studies that report on the AF diagnostic accuracy of digital technologies have been published since the search was conducted for these two existing systematic reviews. Thus, an updated comprehensive review on the diagnostic accuracy of all digital technologies compared to the gold standard will aid in the selection of which digital device(s) can and should be tested in LMICs.” (lines 84-111)

To address point b: We do not plan to pool together all the devices in our review as mentioned in the analysis section of the manuscript: “Where a meta-analysis is possible for any group of primary studies with the same index test (technology), the sensitivity and specificity will be calculated using a random effect model and pooled effects will be presented within the forest plot of individual study estimates”.

Comment 3: In addition, the authors cited a series of systematic reviews and meta-analyses in their references 10 to 14. What are the gaps among these reviews? What else are you adding to the literature? The authors should explain how their proposed review addresses the gaps in the current literature.

Response 3: The proposed review addresses the following gaps: Majority of systematic reviews assessed the diagnostic accuracy of a single technology or two technologies in their review. Indeed, few systematic reviews assessed the diagnostic accuracy of available digital technologies, however there are few imitations such limitation of publication year, and did not restrict to studies using the gold standard as a comparator device (including several studies where the comparator was not described). We have explained this in the introduction as follows:

“Eight existing systematic reviews have assessed the diagnostic accuracy of digital technologies such as PPG, mobile applications, single lead ECGs and smart wearable devices for AF (12–20). However, six of these reviews only assessed one type of technology. For instance, Belani et al (2021), Gill S et al (2022), Nazarian S (2020) et al and Narut et al (2021) only reviewed the diagnostic accuracy of smart gadgets and wearable devices such smart watches (12, 13, 17, 18), and Wong et al (2020) only reviewed diagnostic accuracy of handheld electrocardiogram (20); and O'Sullivan JW (2020) et al reviewed diagnostic accuracy of smartphone camera applications (16). Including all types of technologies in one review, with pooled diagnostic accuracy stratified by type of device would be useful in deciding which device to take forward for testing in new settings, particular in resource-limited settings. Giebel et al reviewed the diagnostic accuracy of all types of technologies; however, the inclusion criteria were restricted to publication years 2014 to 2019, they did not restrict to studies that only used a 12-lead ECG (the gold standard) as the comparison device. Using different comparison devices can lead to discrepancies in diagnostic accuracy. Lopez et al also reviewed all types of technologies, but also limited their search to 2012 to 2019 and did not restrict to studies using the gold standard as a comparator device (including several studies where the comparator was not described). Furthermore, several studies that report on the AF diagnostic accuracy of digital technologies have been published since the search was conducted for these two existing systematic reviews.” (lines 84 - 111)

Comment 4: Two reviewers commented about the lack of clarity in inclusion and exclusion criteria. The authors referred them to their existing statements in the manuscript. Perhaps the authors assume inclusion and exclusion criteria are mutually exclusive. However, the authors may think about other exclusion criteria, such as “studies that involved more than one person in interpreting the same ECG findings without reporting the consensus results among the interpreters.”

Response 4: The existing exclusion criteria is provided in Table 1 (please see below); however, we have reviewed it again in light of your comment and have made some minor changes to it. We also recognised that the exclusion criteria were not specifically mentioned within the text; therefore, we have amended the text accordingly (please see below). Thank you for your suggestion regarding consensus results. After discussing with co-authors, we agreed that we will include studies that may or may not report consensus results; however, this is something we will consider when assessing the risk of bias using the QUADAS-2 tool. 

Table 1: PICOS of the systematic review 

PICOS Inclusion criteria Exclusion criteria 

Population Adults aged 18 years or above 

Intervention Mobile application, ECG based digital technology or non-ECG-based digital technology 

Comparison 12 lead ECG 

Outcome Sensitivity and specificity, the diagnostic odds ratio (DOR), negative and positive predictive values (NPV and PPV), or true positive (TP), true negative (TN), false positive (FP) and false negative (FN) values for detecting atrial fibrillation Studies that do not reporting any of these measures for atrial fibrillation will be excluded 

Study design Cohort and cross-sectional study designs Interventional studies, review articles, editorials, case reports, modelling, economic studies, or clinical trials 

“We will exclude any studies that do not report the outcomes of interest specifically for AF and the following study designs: interventional studies, review articles, editorials, case reports, modelling, economic studies, or clinical trials.” (lines 149-151)

Comment 5: In the second comment, Reviewer 2 asked about the elaboration of the flow of review. The authors replied about including the PRISMA diagram, which is a conventional requirement. I think, in addition to the PRISMA diagram, the authors should elaborate on the number of reviewers in screening titles/ abstracts and how that flows into full-text review (e.g., X number of reviewers check the first N number of full text, resolve discrepancies & reach consensus, then Y number of reviewer proceed with the remaining full-text review).

Response 5: We agree the number of reviewers is important for screening, reviewing of articles and data extraction. This information is already included within the methods section, please see below. Please note this information is also available in the abstract. 

“Two reviewers (AS and SK) will independently review titles and abstracts and the full-text of any study where eligibility cannot be determined via the title and abstract. Disagreements will be handled through discussion or a third reviewer (UP).” (lines 162-165)

“Data will be extracted by two reviewers (AS and SK) independently and cross checked for accuracy.” (lines 168-169)

“The Quality Assessment of Diagnostic Accuracy Studies tool (QUADAS-2) will be used to assess risk of bias independently by two reviewers (AS and SK).” (lines 183-184)

Comment 6: The authors should be aware of potential spectrum effects that have been reported in the literature, i.e., variations of diagnostic accuracy due to differences in the settings in which the devices were used, such as in the community versus the hospital and characteristics of the users. A systematic review and meta-analysis of diagnostic accuracy of handheld ECG devices in diagnosing AF in community and hospital settings reported the variations in AF diagnostic accuracy (doi: 10.1136/heartjnl-2020-316611). Hence, the authors may consider adding the parameter "setting of application" in their review and explore the potential spectrum effect in their reviews.

Response 6: We agree that the setting is extremely important. We have now included within our data extraction methods that we will extract data with regards to setting of device application (grouped under study design/methodology). Additionally, we have also amended our analysis section to include the assessment of setting in terms of application (and not just setting in terms of country-income), please see below.

“We will present the outcomes of sensitivity and specificity using forest plots and will group the studies based on index test first, then by diagnostic threshold of the index test, then by country setting (high-income or low/middle-income) and setting of application (community or healthcare facility).” (lines 199 - 200)

“If heterogeneity is suspected, a Chi-squared test will be performed. Heterogeneity based on sample size (≤100 participants and >100 participants), sampling method (consecutive/convenience), country setting (high-income vs low/middle-income country) and setting of application (community and healthcare facility) will be explored if necessary. If heterogeneity exists but cannot be explained, a meta-analysis will not be performed and results from included studies will be qualitatively synthesized.” (lines 204-211)

---

## [Decision Letter · Decision Letter 2]

21 Mar 2024

Diagnostic accuracy of digital technologies compared with 12-lead ECG in the diagnosis of atrial fibrillation in adults: A protocol for a systematic review

PONE-D-23-03001R2

Dear Dr. Kumarendran,

We’re pleased to inform you that your manuscript has been judged scientifically suitable for publication and will be formally accepted for publication once it meets all outstanding technical requirements.

An invoice for payment will follow shortly after the formal acceptance. To ensure an efficient process, please log into Editorial Manager at Editorial Manager® , click the 'Update My Information' link at the top of the page, and double check that your user information is up-to-date. If you have any billing related questions, please contact our Author Billing department directly at authorbilling@plos.org.

Kind regards,

Ernesto Iadanza

Academic Editor

PLOS ONE

Additional Editor Comments (optional):

Reviewers' comments:

Reviewer's Responses to Questions

**Comments to the Author**

1. Does the manuscript provide a valid rationale for the proposed study, with clearly identified and justified research questions?

Reviewer #2: Partly

Reviewer #5: Yes

2. Is the protocol technically sound and planned in a manner that will lead to a meaningful outcome and allow testing the stated hypotheses?

Reviewer #2: Partly

Reviewer #5: Yes

3. Is the methodology feasible and described in sufficient detail to allow the work to be replicable?

Reviewer #2: Yes

Reviewer #5: Yes

4. Have the authors described where all data underlying the findings will be made available when the study is complete?

Reviewer #2: No

Reviewer #5: Yes

5. Is the manuscript presented in an intelligible fashion and written in standard English?

Reviewer #2: Yes

Reviewer #5: Yes

6. Review Comments to the Author

You may also provide optional suggestions and comments to authors that they might find helpful in planning their study.

Reviewer #2: 1- Discussed and highlight properly the expected limitations and challenges of the current protocol.

2- Present carefully your measures to avoid significant bias, heterogeneity.

Kindest regards

Reviewer #5: Thank you for addressing the comments. The manuscript has addressed its objectives and scope systematically and appropriately.

7. PLOS authors have the option to publish the peer review history of their article (what does this mean?). If published, this will include your full peer review and any attached files.

Reviewer #2: **Yes: **Rami Riziq Yousef Abumuaileq

Reviewer #5: **Yes: **Kam Cheong Wong

---

## [Editor Report · Acceptance letter]

26 Apr 2024

PONE-D-23-03001R2 

PLOS ONE

Dear Dr. Kumarendran, 

I'm pleased to inform you that your manuscript has been deemed suitable for publication in PLOS ONE. Congratulations! Your manuscript is now being handed over to our production team.

Kind regards, 

on behalf of

Dr. Ernesto Iadanza 

Academic Editor

PLOS ONE